# Learning to Correction: Explainable Feedback Generation for Visual Commonsense Reasoning Distractor

### Jiali Chen
South China University of Technology
Guangzhou, China
segarychen@mail.scut.edu.cn

### Xusen Hei
South China University of Technology
Guangzhou, China
202166200175@mail.scut.edu.cn

### Yuqi Xue
South China University of Technology
Guangzhou, China
202130461885@mail.scut.edu.cn

### Yuancheng Wei
South China University of Technology
Guangzhou, China
202130560779@mail.scut.edu.cn

### Jiayuan Xie
The Hong Kong Polytechnic
University
Hong Kong, China
jiayuan.xie@polyu.edu.hk

### Yi Cai[*]
South China University of Technology
Guangzhou, China
ycai@scut.edu.cn

### Qing Li
The Hong Kong Polytechnic
University
Hong Kong, China
csqli@comp.polyu.edu.hk

## Abstract

Large multimodal models (LMMs) have shown remarkable performance in the visual commonsense reasoning (VCR) task, which aims to answer a multiple-choice question based on visual commonsense within an image. However, the ability of LMMs to correct potential visual commonsense errors in the distractor upon their occurrence is yet under-explored. Drawing inspiration from how a human teacher crafts challenging distractors to test students' comprehension of the concepts or skills and assists them in identifying and correcting errors toward the answer, we are the pioneering research for LMMs to simulate this error correction process. To this end, we employ GPT-4 as a "teacher" to collect the explainable feedback dataset VCR-DF for error correction, which serves as a benchmark to evaluate the ability of LMMs to identify misconceptions and clarify reasons behind the error in VCR distractors toward final answers. In addition, we propose an LMM-based Pedagogical Expert Instructed Feedback Generation (PEIFG) model to incorporate the learnable expert prompts and multimodal instruction as guidance for feedback generation. Experimental results show that our PEIFG significantly outperforms existing LMMs. We believe that our benchmark provides a new direction for evaluating the capabilities of LMMs. [1]

[*]Corresponding author.

[1]Code is available at https://github.com/Gary-code/PEIFG

## CCS Concepts

• **Computing methodologies → Natural language generation**; **Scene understanding**.

## Keywords

Large Multimodal Model, Visual Commonsense, Error Correction

**ACM Reference Format:**
Jiali Chen, Xusen Hei, Yuqi Xue, Yuancheng Wei, Jiayuan Xie, Yi Cai, and Qing Li. 2024. Learning to Correction: Explainable Feedback Generation for Visual Commonsense Reasoning Distractor. In *Proceedings of the 32nd ACM International Conference on Multimedia (MM '24), October 28-November 1, 2024, Melbourne, VIC, Australia.* ACM, New York, NY, USA, 10 pages. https://doi.org/10.1145/3664647.3681590

## 1 Introduction

Visual commonsense reasoning (VCR) task aims to predict the answer to the multiple-choice question and provide a convincing rationale [11, 24, 44, 45, 49] about the image. In recent years, it has gained considerable attention from computer vision (CV) and natural language processing (NLP) communities due to the advancement of large multimodal models (LMMs) [3, 14, 24, 41, 46, 51]. Specifically, inferring a reliable answer in VCR requires LMMs to not only recognize objects and scenes but also deeply understand the underlying visual commonsense (e.g., likely intents, goals, and social dynamics of people) in the image.

Scrutinizing existing LMMs [3, 14, 24, 51], we identify their common paradigm as two stages: pre-training and instruction tuning with a large language model. Specifically, they are first pre-trained on large-scale image-text pairs for modality alignment and then construct instruction data that combines both visual and language features, which is fed into the large language model to infer the answer. By this training paradigm, they often demonstrate strong reasoning and generalization capabilities attributed to the exponential increase in both data size and model scale. In the field of

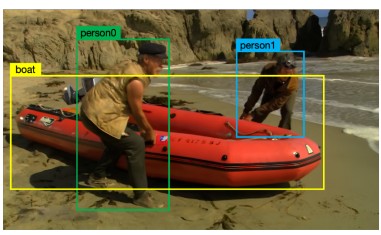

**Question**: What are person0 and person1 doing?

**Answer:** They are moving the boat into the ocean to escape.

**Rationale**: It looks like they are in a hurry to move the boat into the water and escape from something.

**Previous distractors**:
A. Person0 and person1 are having a playdate.
B. Person0 and person1 are observing the test.
C. Person0 and person1 are racing go - karts against each other.

**Our Distractors**:
A. Person0 and person1 are playing a game of tug-of-war with the boat.
B. Person0 and person1 are trying to pull the boat out of the water.
C. Person0 and person1 are setting up a boat display on the beach.

**Our Feedback**:
**Educational level:** Understand

A.
  **Misconception:** Misinterpretation of the activity's purpose.
  **Explanation:** The distractor fails to understand urgency and direction of the action, incorrectly assuming it's a game rather than an effort to move the boat into the ocean.

B.
  **Misconception** : Incorrect direction of the boat movement.
  **Explanation:** The distractor arises from focusing on the act of pulling without considering the context of urgency and the goal to escape, leading to the false conclusion that they are trying to pull the boat out of the water.

C.
  **Misconception** : Misconception of the scene's context.
  **Explanation:** The distractor mistakenly associated information points in the scene, thinking that they are setting up a boat display on the beach, ignoring signs of their eagerness to move the boat into the water to escape.

**Figure 1: A sample of original VCR and our VCR-DF datasets.**

pedagogy [7, 31], a human teacher often designs challenging distractors to evaluate students' cognitive skills in multiple-choice questions and guide them in identifying and rectifying potential errors toward correct answers. While the capability of LMMs for straightforward reasoning has advanced, we further investigate whether LMMs can emulate the error correction process, which remains unexplored in previous LMM studies. Therefore, we generate feedback as error correction based on the multiple-choice question of VCR, which mainly includes the misconception and explanation about the distractor. However, distractors from the original VCR dataset [49] are inadequate for evaluating the error correction ability of LMMs since the inherent bias in these distractors. Specifically, the sample from the original VCR dataset frequently exhibits a higher overlap between the correct answer and entities than with distractors, where merely choosing the option based on the rule of maximum entity overlap can achieve over 60% accuracy. Moreover, the content of the distractors frequently lacks any pertinent relevance to the image in the original VCR dataset. As shown in Fig. 1, elements "playdate", "test" and "racing" from VCR dataset have no connection to the image. Consequently, LMMs may easily dismiss these distractors without comprehensively understanding the visual commonsense, leading to a lack of visual commonsense errors for feedback generation.

To address the aforementioned issues, we construct the VCR-DF dataset including new distractors and explainable feedback, which can serve as a benchmark to evaluate the error correction ability

of LMMs. Specifically, we first utilize the language-only GPT-4 to classify the educational level of the current question with Bloom's taxonomy [2], which can be categorized into "remember", "understand", "apply", "analyze", "evaluate" and "create" levels. This classification prevents the subsequent generation of distractors from deviating from the corresponding educational level. Next, following [27], given the input information (i.e., event, question, answer, and objects) and educational level, GPT-4 generates new distractors related to the content of the image, which are subject to further manual screening. Finally, we employ GPT-4 again as a "teacher" to annotate the corresponding misconception and explanation about the distractor. As shown in Fig. 1, the distractors from our VCR-DF contain visual commonsense errors, which are more relevant to the image. The feedback from VCR-DF includes the educational level, misconception and explanation of the distractor.

In this paper, we propose a feedback generation task for VCR distractors to evaluate the error correction ability of LMMs. Diverging from the forward reasoning, LMMs often overlook the potential mismatched image-text information between the distractor and image since they typically learn from image-text pairs during training. Moreover, data specific to error correction is often absent in pre-training datasets, resulting in models' inability to rectify errors when encountered. Experimental results also show that LMMs do not exhibit the same proficiency in error correction as they do in reasoning. Therefore, we argue that equipping LMMs with strong reasoning abilities to perform error correction requires specialized prompts as guidance of the LMM. Building upon this analysis, we introduce Pedagogical Expert Instructed Feedback Generation (PEIFG), an LMM engineered for error correction by feedback generation, which consists of three key components: a visual feature extractor (VFE), an expert prompt selector (EPS), and a text generator. Specifically, the VFE first utilizes the visual marker perceiver (VMP) and CLIP [34] image encoder to extract region-level and global-level visual features, which are concatenated as the image representation. To address the challenge of aligning textual information with multiple instances of the same object category (e.g., person0 and person1) within the image, we add object boxes as visual markers for corresponding objects and then incorporate the SAM-based [20] VMP with a language model (i.e., OPT-350M [50]) for object coordinates prediction. Subsequently, we develop the EPS to select expert prompts from a learnable prompt pool, representing specialized expert knowledge as guidance for feedback generation. Technically, we combine language instructions with image representations to select the most relevant expert prompts. Finally, we integrate the text prompt, visual features from VFE and expert prompts into the multimodal instruction, which guides the large language model for feedback generation. Furthermore, a refinement step employs reinforcement learning to ensure the generated feedback is explainable and faithful, maintaining logical consistency with the input information. Considering our model effectively identifies visual commonsense errors within distractors, we can further utilize PEIFG to capture these potential errors and generate distractors without modifying the model architecture.

Our main contributions can be summarized as follows:

- To the best of our knowledge, we are the first to investigate the error correction capabilities of large multimodal models (LMMs).

Additionally, we construct a benchmark and introduce the feedback generation task for evaluation.

- To engage the LMM with strong reasoning abilities in error correction, we propose the Pedagogical Expert Instructed Feedback Generation (PEIFG) model, which effectively integrates visual features and learnable expert prompts into the multimodal instruction for feedback generation.
- Extensive experiments on our benchmark show the proposed PEIFG model significantly surpasses the existing LMMs, and offers a new direction to evaluate the capabilities of LMMs.

## 2  Related Work

Equipping machines with multimodal reasoning ability is a long-standing goal of artificial intelligence (AI) systems [28, 48, 53]. This form of reasoning empowers machines to emulate human-like cognition and commonsense understanding of the world. Recent progress in multimodal learning has been driven by incorporating visual features with pre-trained large language models as large multimodal models (LMMs) [8, 17, 27, 54]. The current paradigm for training LMMs primarily comprises two stages (image-text pre-training and instruction tuning). Specifically, during the image-text pre-training stage, these LMMs are initially trained on a large scale of image-text pairs for cross-modal alignment [1, 21, 22, 47]. This process ensures that both visual and textual input information are effectively mapped into a unified semantic space. For example, BLIP-2 [22] design a Q-Former to align visual and textual features in the pre-trained stage. During instruction tuning, LMMs fine-tune on multimodal instruction datasets to enhance their instruction-following ability and tackle more complex multimodal tasks. These instruction datasets originate from manually annotated data [5, 9, 15, 30, 38] and data generated by GPT-4 [4, 23, 26, 27, 47]. InstructBLIP [8] builds upon the BLIP-2 framework by fine-tuning Q-Former with instruction data. MiniGPT-4 [54] directly employs a linear layer to project visual features into the semantic space of language, leveraging instruction data for this process. LLaVA [27] is the first LMM to fine-tune through self-instruction, which employs language-only GPT-4 to generate instruction-following data with some manual samples for fine-tuning. Hong et al. [17] propose an 18B LMM CogAgent for graphical user interfaces (GUI) understanding and navigation, which utilizes both low-resolution and high-resolution image encoders. Meanwhile, it achieves the state-of-the-art on various reasoning benchmarks. Moreover, some other works [6, 16, 29, 39] use the large language models as the controller, which aims to control various visual modules with code generation for reasoning. Specifically, VisProg [16] and ViperGPT [39] utilize predefined APIs to access available modules, and compose them by generating Python code for execution without any task-specific training. Existing LMMs primarily focus on forward reasoning capabilities in multimodal tasks. However, their ability to analyze the causes of errors and rectify them is yet under-explored. To the best of our knowledge, we are the first to investigate LMMs for error correction in the visual commonsense reasoning task.

## 3  Dataset Construction

Existing datasets for visual commonsense reasoning (VCR) [11, 49] are mainly structured as multiple-choice questions. However, they are inadequate for comprehensively assessing models' visual commonsense reasoning capability and their proficiency in error correction. The reasons are as follows: i) The original distractors within these datasets contain inherent bias, where excessive overlap between the entities of the image and answer, and the distractors often lack relevance to the question and image. It results in a deficiency of visual commonsense errors within these distractors. ii) They are deficient in feedback mechanisms to identify and address errors within distractors, which is essential for timely correction. Inspired by cognitive-developmental theory in the field of pedagogy [31], teachers often employ Bloom's taxonomy [2] to identify questions at various levels and then craft corresponding distracors, aiming to probe students' potential cognitive challenges. Moreover, they provide feedback to assist students in understanding and correcting their errors after failing to solve a problem. Therefore, we construct the VCR-DF dataset as a benchmark for evaluating the error correction ability of large multimodal models (LMMs) in visual commonsense reasoning. In total, the VCR-DF dataset contains 22,401 data samples and splits 20,163 and 2,238 samples for training and testing respectively.

Specifically, our VCR-DF dataset is derived from the original VCR dataset [49], which leverages GPT-4 [32] for the new distractors and feedback data collection. For the input image $I$, question $Q$, and correct answer $A$, we prompt GPT-4 to generate multiple distractors $\{D_i\}_{i=1}^{N}$ and corresponding feedback $\{F_i\}_{i=1}^{N}$. The prompts for GPT-4 are shown in the Supplementary. In the following subsection, we will detail the procedure of data collection.

### 3.1  Distractor and Feedback Collection

To avoid the laborious demand of human annotation, we design a data reformation pipeline assisted by language-only GPT-4 and a manual filtering process for distractor and feedback collection. Specifically, the source images are from the original VCR dataset [49]. We also preserve questions, correct answers and object boxes provided in VCR.

*3.1.1  **Distractor Collection**.* For distractor data collection, we guide GPT-4 to identify the Bloom's taxonomy level of the question with the given answer, and generate five distractors with the corresponding Bloom's taxonomy level for each QA pair based on the given inputs (i.e., manually annotated image events and places, questions, answers, and object boxes).

Considering the potential mistakes in annotations with GPT-4, we develop a web page for manual distractor filtering to remove those of lower quality. We first ask a group of trained annotators to assess whether each distractor is related to the question and image. Upon confirming relevance, they need to review the distractor against the image, answer, and question for inaccuracies, where distractors without errors are discarded. Finally, annotators rank the distractors and select the top-3 of them.

*3.1.2  **Feedback Collection**.* Given the above input information and each filtered distractor, we instruct GPT-4 to identify the misconception and explain the error in the distractor, serving as the preliminary feedback data. Subsequently, we manually classify feedback meeting the following quality criteria as the final feedback data. (i) **Accuracy**: must precisely pinpoint and correct the error

within the distractor, ensuring the rectification is directly relevant to the identified mistake. (ii) **Clarity**: The explanation provided in the feedback should be clear and understandable, avoiding ambiguity or overly complex language that could hinder comprehension. After our manual evaluation, over 90% of the preliminary samples are both meet the above two criteria. Finally, we integrate synthetic Bloom's taxonomy levels, misconceptions and explanations about the error as feedback data, as shown in Fig. 1.

## 4 Methodology

Our goal is to develop a large multimodal model (LMM) for explainable feedback generation, which simulates the teaching process of educators. Specifically, given the input image $I$, question $Q$, correct answer $A$, and $N$ distractors $\{D_i\}_{i=1}^{N}$, we identify the corresponding Bloom's taxonomy level of the question, and then generate the misconception and explanation as feedback $\{F_i\}_{i=1}^{N}$. The overall architecture of our proposed Pedagogical Expert Instructed Feedback Generation (PEIFG) model is shown in Fig. 2, which consists of three components: (i) visual feature extractor (VFE), which adds the object box information as visual markers into the image and obtains the contextually enriched visual features. (ii) expert prompt selector (EPS), which incorporates language instructions and visual features to select the most relevant expert prompts as expert knowledge from a learnable prompt pool. (iii) text generator, which constructs a multimodal instruction for the large language model (LLM), integrating the visual features, expert prompts and language instruction to generate the feedback. The details of each component are shown in the following subsections.

### 4.1 Visual Feature Extractor

Different from previous visual question answering datasets (e.g., VQA v2.0, OKVQA and so on), where the input is an image and question. The VCR dataset provides the model with additional object boxes including multiple instances of the same object category (e.g., person0 and person1). Consequently, we first automatically annotate the objects within the image with the provided boxes as visual markers, as shown in Fig. 2. Then, we introduce a visual marker perceiver (VMP) to comprehend the visual markers information (i.e., object boxes and object text annotation) in the image by stage 1 training, as shown in Fig. 2. The region-level features from the VMP and global-level features from CLIP [34] image encoder are concatenated as final visual features.

#### 4.1.1 *Visual Marker Perceiver*. 
Given the image with visual markers, we enable the visual marker perceiver (VMP) to understand the object with markers in stage 1 training. Specifically, we initially reshape the image into high resolution (i.e., $1024 \times 1024$) and use the SAM-base [20] backbone with two convolution layers as VMP to obtain the region-level visual features.

Following [43], we fed the region-level features and detection instruction into a language model (i.e., OPT-350M) [50] to effectively enhance the VMP capability in discerning object spatial information through visual markers. Technically, we employ a multilayer perception (MLP) to map region-level visual features into the semantic space of language. For the input of OPT model, we fill them into the pre-defined detection instruction template, i.e., " Detect all objects in the image". The special token "" is replaced

by the region-level visual features. The coordinates of the object boxes are provided for prediction by the OPT model. Specifically, we normalize the coordinates of top-left and bottom-right corners to the range of $[0, 1]$ according to the image size, which provides the OPT [50] model for prediction. The language modeling loss is utilized to optimize both the VMP and OPT models concurrently, facilitating object perception with visual markers.

#### 4.1.2 *Visual Features Extraction*. 
After stage 1 training, we employ the trained VMP and CLIP image encoder to obtain region-level and global-level features, respectively. Specifically, given the high-resolution image with visual markers $I_h$, VMP extracts the region-level visual features. Meanwhile, we utilize the CLIP image encoder [34] to encode the low-resolution image $I_r \in \mathbb{R}^{224 \times 224}$, and then obtain the global-level visual features. Subsequently, we map both types of visual features into the semantic space of language, enhancing their compatibility with the input format of the large language model for further feedback generation, denoted as $v_r$ and $v_g$ respectively. The process of visual feature extraction can be formulated as:

$$
\begin{aligned}
v_r &= \text{MLP}_r(f_r(I_h)), \\
v_g &= \text{MLP}_g(f_g(I_r)),
\end{aligned}
\tag{1}
$$

where $f_r$ and $f_g$ represent the VMP module and CLIP image encoder. $\text{MLP}_r$ and $\text{MLP}_g$ are multilayer perception (MLP) layers. $v_r \in \mathbb{R}^{L_d \times d}$ and $v_g \in \mathbb{R}^{L_d \times d}$. $L_d$ is the length of image patches and the dimension of features $d$ is 1024. Finally, we concatenate $v_r$ and $v_g$ as the integrated visual features, denoted by $v = [v_r; v_g]$, where $v \in \mathbb{R}^{L_d \times 2d}$ and $[;]$ is the concatenation operation.

### 4.2 Expert Prompt Selector

The introduction of the expert prompt selector (EPS) is motivated by the desire to enable our model to emulate the diverse expertise of teachers as expert knowledge in generating feedback. Thus, we maintain a learnable prompt pool representing diverse expert knowledge and employ instruction-aware visual features (i.e., integrating both visual and language instruction) to select the relevant expert prompts from the pool.

#### 4.2.1 *Instruction-aware Visual Feature*. 
We design manually crafted natural language instructions to align with the integrated visual features for further expert prompt selection. Additionally, we supplement these instructions with the corresponding question and answer as auxiliary information. Following [8], we use the Query Transformer (Q-Former) [22] module to extract the instruction-aware visual features. Technically, within the Q-Former, learnable query tokens align with the language instructions by self-attention mechanisms and align with the integrated visual features by cross-attention mechanisms. Next, the output sequence of Q-Former is fed into an average-pooling layer to obtain instruction-aware visual features $v_s$. The computation of this process can be expressed as:

$$
v_s = \text{Avg}(\text{Q-Former}(X_q, X_n, v)),
\tag{2}
$$

where $v_s \in \mathbb{R}^{768}$. $X_q \in \mathbb{R}^{L_q \times 768}$ is the learnable query tokens and $X_n \in \mathbb{R}^{L_n \times 768}$ denotes embeddings of the manually crafted instruction. $L_q$ and $L_n$ correspond to the lengths of the query tokens and instruction, respectively. Q-Former$(\cdot)$ and Avg$(\cdot)$ represent the Q-Fromer module and average-pooling layer respectively.

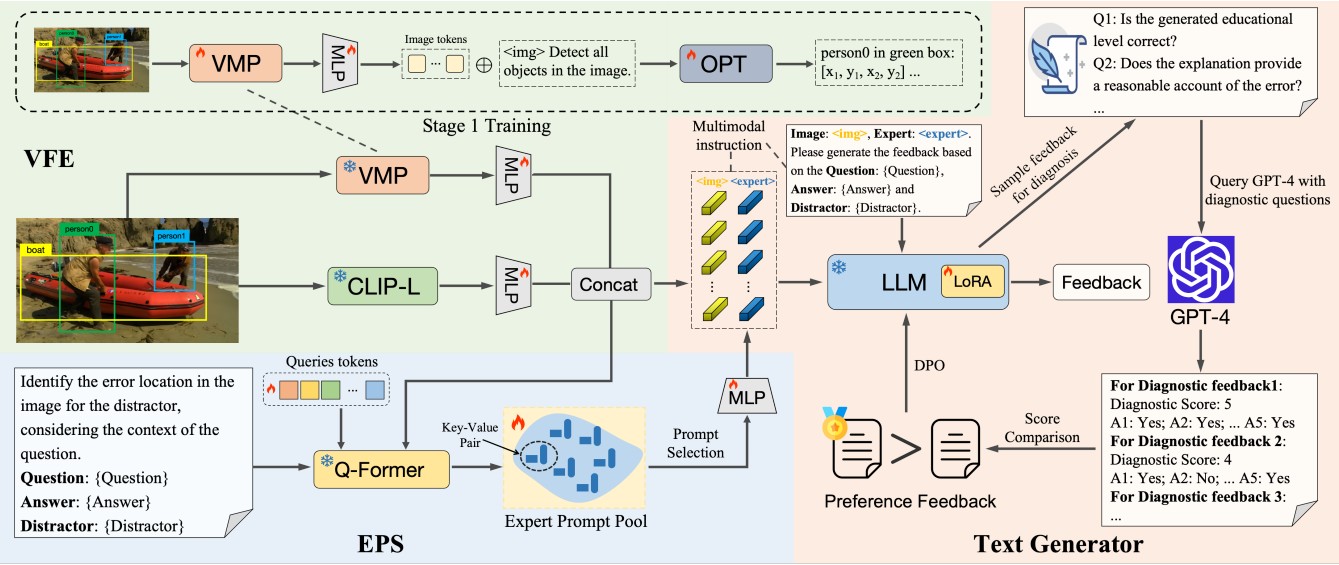

**Figure 2: Overview of our PEIFG model. It contains three components: (i) the visual feature extractor (VFE), (ii) the expert prompt selector (EPS), (iii) the text generator. `` and `<expert>` indicate the integrated visual features and top-$K$ selected expert prompts, respectively.**

*4.2.2* ***Expert Prompt Selection.*** To select the expert prompts relevant to the current input information (i.e., the image, question and answer), we use the instruction-aware visual features as guidance for selection. We first build a learnable prompt pool to maintain diverse expert knowledge which can be formulated as:

$$\mathcal{P} = \{P_1, P_2, ..., P_S\}, \qquad (3)$$

where $P_i \in \mathbb{R}^{L_p \times 768}$ and $L_p$ is the length of each expert prompt. $S$ denotes the size of the prompt pool. Given the necessity for the pool to encompass a wide array of expert prompts, we consider that each prompt should retains its distinct expertise. Each prompt should be jointly independent in the sense that each of them contains unique information. To this end, we design the expert correlation loss to reduce the correlation among the expert prompts in the pool by minimizing their inner product:

$$\mathcal{L}_{cor} = \left\| \mathcal{P}\mathcal{P}^T - \text{diag}\left(\mathcal{P}\mathcal{P}^T\right) \right\|_F^2, \qquad (4)$$

where $\text{diag}(\cdot)$ only preserves diagonal entries and $|| \cdot ||_F^2$ is the square of the Frobenius norm. Subsequently, we employ the query and match key-value based strategy for expert prompt selection. Concretely, we utilize the average pooling result of each expert prompt as its corresponding key, denoted as $\{k_1, k_2, ..., k_S\}$, where $k_i \in \mathbb{R}^{768}$. Importantly, the keys are updated corresponding to the expert prompts during training. Given the input information, we aim to find out the top-$K$ keys $\{k_{m_1}, k_{m_2}, ..., k_{m_K}\}$, making them closer to the input sample, where $\{m_j\}_{j=1}^K$ denotes a subset of $K$ ($1 \le K \le S$) indices of keys in the pool. Specifically, we calculate the cosine similarity $sim(\cdot)$ between keys and the instruction-aware visual features $v_s$ to select top-$K$ keys and calculate the key matching

loss for selection:

$$\bar{k} = \underset{\{m_i\}_{i=1}^K \subseteq [1,S]}{\text{argmin}} \sum_{i=1}^K sim\left(v_s, k_{m_i}\right), \qquad (5)$$

$$\mathcal{L}_{se} = -\sum_{\bar{k}} sim(k_{m_i}, v_s), \qquad (6)$$

where $\bar{k}$ is the set of top-$K$ keys. Finally, we obtain the top-$K$ expert prompts from the pool corresponding to the selected keys:

$$\hat{P} = [P_{m_1}; P_{m_2}; ...; P_{m_K}], \qquad (7)$$

where $\hat{P} \in \mathbb{R}^{(K \times L_p) \times 768}$, $m_i$ denotes indices of selected prompts $\hat{P}$ and $[;]$ is the concatenation operation.

### 4.3 Text Generator

Upon acquiring the integrated visual features and top-$K$ expert prompts, we fuse them into the LLM-based text generator (i.e., QWen1.5) for feedback generation. Specifically, we adopt the widely used instruction tuning method to incorporate the language prompt, integrated visual features, and expert prompts into multimodal instruction. We first utilize a multilayer perception (MLP) to project the expert prompts into a $2d$ dimensional space, meeting the input requirement of the LLM. The multimodal instruction is defined as: "Image: ``. Expert: `<expert>`. Please generate the feedback based on the question: {Question}, answer: {Answer}, distractor: {Distractor}". The special tokens "``" and "`<expert>`" are replaced by integrated visual features and selected expert prompts, respectively. "{Question}", "{Answer}" and "{Distractor}" are the input question, answer and distractor of a specific sample. The multimodal instruction is directly fed into the frozen large language model with learnable LoRA layers [18] for language modeling. The

cross-entropy loss of language modeling can be formulated as:

$$\mathcal{L}_{lan} = - \sum_{t=1}^{T} \log p_\theta \left( w_t \mid w_{<t}, \text{Ins} \right), \tag{8}$$

where $\log p_\theta(\cdot)$ is the negative log-likehood, Ins is the multimodal instruction, and $w_{<t}$ represents the words before the $t$-th word.

Given our method's proficiency in identifying visual commonsense errors in distractors, we can utilize the PEIFG model to further capture these potential errors and generate distractors.

*4.3.1  **Refinement**. To ensure the logical coherence between the feedback and input information, we leverage the trained PEIFG model to generate pseudo training feedback data, which is then used to refine the model performance with the assistance of GPT-4. Specifically, we randomly sample from the VCR-DF training data and employ the top-p sampling method to generate $M$ feedback instances for each sample, denoted as $\{\hat{F}_1, \hat{F}_2, ..., \hat{F}_M\}$. Given the question, ground truth distractor, feedback, and the generated feedback, we formulate five diagnostic questions for GPT-4 to ascertain whether the generated feedback meets the specified criteria. For each diagnostic question, feedback that meets the criteria is awarded 1 point, otherwise 0 point. Therefore, the final diagnostic score $s_d$ for each generated feedback ranges from 0 to 5. Finally, we rank the generated feedback as pairs based on the diagnostic score and design a refinement loss $\mathcal{L}_{re}$, which utilizes the direct preference optimization (DPO) [36] as reinforcement learning algorithm to further optimize the LLM. More details about the refinement process are shown in the Supplementary.

## 4.4  Training Objective

In our setting, we treat the generation of feedback as two distinct tasks. Specifically, feedback generation, we generate feedback for error correction for each given distractor within the VCR-DF dataset. The objective of our PEIFG model is to minimize the total loss, i.e., key matching loss in Eq. 6, correlation loss in Eq. 4 and language modeling loss in Eq. 8. The definition of total loss is:

$$\mathcal{L} = \frac{1}{D} \sum_{t=1}^{D} (\mathcal{L}_{lan} + \lambda_1 \mathcal{L}_{cor} + \lambda_2 \mathcal{L}_{se}), \tag{9}$$

where $D$ is the total number of training samples, $\lambda_1$ and $\lambda_2$ stand for hyperparameters. It is worth noting that the refinement loss $\mathcal{L}_{re}$ is applied to optimize the model's parameters after the PEIFG has been trained.

## 5  Experiment

### 5.1  Implementation Details

We implement our PEIFG model with Pytorch and train it on two RTX 3090 cards. For the visual feature extraction, we employ the SAM-base [20] backbone and two convolution layers as the visual marker perceiver (VMP). Furthermore, in this stage 1 training for the VMP module, we use AdanW [19] optimizer with an initial learning rate of 5e-5. The ViT-L/14 [12] pre-trained in CLIP [34] is used for the image encoder, where ViT-L/14 denotes ViT-Large model with the patch size $14 \times 14$. Thus, the length of image tokens $L_d = 256$ and the dimension of features $d$ is 1024. When selecting the expert prompt, we set the size of the prompt pool $S$ to 10 and

the length of each expert prompt $L_d$ to 5. The number of selected prompts $K$ is 3. The application of expert prompts differs in distractor and feedback generation. Specifically, we leverage top-3 expert prompts, which allows for a more comprehensive analysis by incorporating diverse expert knowledge into multimodal instruction. For distractor generation, within single forward, we aim to generate 3 distractors. To achieve this, each of the three expert prompts is utilized to generate one corresponding distractor, ensuring a tailored approach to leverage the distinct expertise of each expert prompt. We choose the QWen1.5-1.8B as the LLM for distractor and feedback generation. The LoRA layers are inserted into each self-attention layer of the LLM for optimization. During training, we use Adam optimizer with cosine scheduler and the initial learning rate of 8e-5 to optimize the total loss function $\mathcal{L}$. We only fine-tune the learnable query tokens of Q-Former, expert prompt pool, multilayer perception (MLP) and LoRA layers of LLM, while freezing other parameters. For LoRA, we set the rank to 8. The batch size is 24 and we train 3 epochs. The hyperparameters $\lambda_1$ and $\lambda_2$ for the total loss function in Eq. 9 are both 0.1.

### 5.2  Baseline and Ablation Models

*5.2.1  **Baseline Models**. In this paper, we evaluate our proposed PEIFG by comparing it with two types of baseline models.

- Explanation-Enhanced visual question answering models including NLX-GPT [37] and KICNLE [42]. Specifically, NLX-GPT adopts the pre-trained CLIP encoder and GPT-2 [35] language model for feedback and distractor generation. KICNLE incorporates external knowledge and designs a multi-iteration generative approach to ensure logical consistency between the generated distractor and feedback.
- Multimodal large language models (LMMs) with different parameter scales from 3B to 18B, including BLIP-2 [22], InstructBLIP [8], VisualGLM [13], LLaVA-v1.5 [27] and CogAgent [17]. Specifically, we integrate the LoRA layers into the self-attention mechanisms of LLMs. Furthermore, we randomly select 200 samples for comparison between our model and the GPT-4 with vision (GPT-4V) [32], which boasts parameter scales above 175B and has the best performance on various multimodal tasks.

More implementation details of baseline models are provided in the Supplementary.

*5.2.2  **Ablation Models**. To investigate the performance effect of each module in PEIFG, we compare the following variants of our method on VCR-DF dataset. We independently conduct corresponding ablation experiments for each module.

- **PEIFG w/o stage 1**: PEIFG without stage 1 training for visual marker percevier.
- **PEIFG w/o VMP**: PEIFG without visual marker percevier for visual feature extraction.
- **PEIFG w/o CLIP**: PEIFG without CLIP image encoder for visual feature extraction.
- **PEIFG w/o EPS**: PEIFG without expert prompt selector, which removes expert prompts from the multimodal instruction.
- **PEIFG w/o Ref**: PEIFG without refinement for generation.

**Table 1: Main automatic metrics results of baselines and our model. Bold: the maximum value in the column.**

| Task | Model | BLEU-1 | BLEU-2 | BLEU-3 | BLEU-4 | METEOR | ROUGE$_L$ | CIDEr | BERTScore |
|------|-------|--------|--------|--------|--------|--------|--------|-------|-----------|
| Feedback | NLX-GPT [37] | 42.59 | 25.32 | 16.29 | 10.87 | 33.69 | 32.61 | 3.08 | 63.71 |
| | KICNLE [42] | 41.05 | 24.37 | 14.94 | 10.12 | 33.46 | 32.84 | 4.23 | 63.67 |
| | BLIP-2 [22] | 42.59 | 26.47 | 18.15 | 13.88 | 33.25 | 39.70 | 17.92 | 69.50 |
| | InstructBLIP [8] | 43.80 | 28.92 | 20.44 | 14.95 | 34.35 | 40.84 | 16.34 | 70.52 |
| | VisualGLM [13] | 44.72 | 28.32 | 19.47 | 13.93 | 30.54 | 37.72 | 19.71 | 70.54 |
| | LLaVA-v1.5 [27] | 43.79 | 26.87 | 17.25 | 10.74 | 29.77 | 35.55 | 21.71 | 68.59 |
| | CogAgent [17] | 46.47 | 30.95 | 20.79 | 13.07 | 33.40 | 38.81 | 31.65 | 70.95 |
| | PEIFG w/o stage 1 | 45.74 | 30.78 | 22.45 | 16.61 | 35.20 | 39.91 | 28.75 | 70.40 |
| | PEIFG w/o VMP | 46.41 | 30.84 | 22.18 | 15.95 | 34.40 | 40.07 | 26.77 | 69.84 |
| | PEIFG w/o CLIP | 46.37 | 31.03 | 22.66 | 16.89 | 34.85 | 40.40 | 25.51 | 70.50 |
| | PEIFG w/o EPS | 44.38 | 29.20 | 22.89 | 15.15 | 34.10 | 39.64 | 26.72 | 70.62 |
| | PEIFG w/o Ref | 45.73 | 30.57 | 22.11 | 16.20 | 34.49 | 39.88 | 29.27 | 70.77 |
| | PEIFG | **46.53** | **31.77** | **23.45** | **17.69** | **35.77** | **41.08** | **32.10** | **71.60** |
| Distractor | NLX-GPT [37] | 10.31 | 7.10 | 5.04 | 3.67 | 20.21 | 31.46 | 2.18 | 43.82 |
| | KICNLE [42] | 10.72 | 7.62 | 5.54 | 4.15 | 20.97 | 33.17 | 5.39 | 43.74 |
| | BLIP-2 [22] | 32.59 | 17.18 | 10.06 | 5.62 | 34.75 | 30.00 | 21.60 | 68.90 |
| | InstructBLIP [8] | 31.94 | 16.82 | 9.84 | 5.49 | 34.51 | 29.92 | 20.19 | 68.62 |
| | VisualGLM [13] | 44.29 | 29.19 | 20.76 | 14.62 | 39.51 | 39.40 | 78.84 | 74.97 |
| | LLaVA-v1.5 [27] | 39.14 | 27.35 | 20.19 | 13.61 | 38.58 | 38.25 | 68.59 | 73.65 |
| | CogAgent [17] | 42.23 | 30.94 | 22.67 | 15.29 | 40.82 | 40.54 | 78.23 | 74.81 |
| | PEIFG w/o stage 1 | 46.13 | 33.03 | 22.71 | 17.09 | 43.84 | 43.49 | 66.62 | 75.12 |
| | PEIFG w/o VMP | 44.90 | 31.77 | 22.49 | 16.68 | 43.12 | 43.49 | 67.34 | 74.69 |
| | PEIFG w/o CLIP | 46.09 | 32.82 | 23.46 | 17.73 | 43.17 | 42.62 | 67.49 | 75.49 |
| | PEIFG w/o EPS | 43.97 | 30.54 | 20.68 | 16.25 | 41.78 | 41.12 | 62.81 | 74.59 |
| | PEIFG | **47.46** | **34.33** | **24.53** | **18.41** | **44.00** | **43.90** | **78.98** | **76.35** |

**Table 2: Comparison of automatic metrics between PEIFG and GPT-4V on 200 randomly sampled feedback.**

| Method | BLEU-4 | METEOR | CIDEr | BERTScore |
|--------|--------|--------|-------|-----------|
| GPT-4V | 7.69 | 33.57 | 24.47 | **71.07** |
| PEIFG | **15.23** | **34.38** | **29.44** | 70.98 |

## 5.3 Evaluation Metric

*5.3.1 **Automatic Evaluation Metrics**.* We evaluate the performance with eight standard metrics, including BLEU-(1 to 4) [33], ROUGE$_L$ [25], METEOR [10], CIDEr [40], and BERTScore [52]. These metrics are commonly used for evaluating text generation and we compute metric values using the publicly available code[2].

## 5.4 Results and Analysis

*5.4.1 **Performance Comparison**.* Table 1 and 2 show the automatic evaluation results of baselines and our model on feedback and distractor generation. We find that: **i)** Experimental results in Table 1 provide evidence that our PEIFG also surpasses the open-source baselines on feedback generation task. Specifically, compared with two explanation-enhanced visual question answering models (i.e., NLX-GPT and KICNLE) trained with full fine-tuning, the feedback generated by them often fails to produce lengthy textual content of the feedback, while merely providing the correct answer. It suggests

that these small models encounter limitations in generating feedback for error correction. By comparing our model with existing LMMs with parameters ranging from 3B to 18B, our performance still surpasses theirs, which indicates that these general-purpose LMMs are not ideally suited for error correction in visual commonsense reasoning without specific prompt information to instruct the LLM. Worse still, VisualGLM and LLaVA-v1.5 exhibit a bias towards predicting the "understand" level for almost all samples, since instances of this level are the most in the training data. Conversely, our model achieves a prediction accuracy of 88% for educational levels. **ii)** Notably, Table 2 shows that our PEIFG model achieves better performance in automatic evaluation metrics when compared with GPT-4V in feedback generation. Additionally, we observe that feedback generated by GPT-4 tends to be overly verbose, which is not conducive to human learning from the feedback and leads to lower N-gram metric scores. **iii)** We also evaluate the quality of generated distractors. As shown in Table 1, our PEIFG model consistently outperforms all baselines with significant margins on all metrics. For example, PEIFG outputforms CogAgent model by margin of "+3.12" and "+1.54" in BLEU-4 and BERTScore scores, respectively. These results indicate that the distractors produced by PEIFG more closely align with the ground truth, which can be attributed to PEIFG's more effective capture of visual commonsense errors within distractors during the feedback generation process.

*5.4.2 **Ablation Study**.* We conduct ablation experiments to verify the effectiveness of different components in our PEIFG model. Experimental results are also shown in Table 1. We observed that: **i)**

---

[2]https://github.com/huggingface/evaluate

By removing the stage 1 training for visual marker perceiver (VMP), we notice an inferior performance on both feedback and distractor generation. It demonstrates that our PEIFG becomes adept at understanding objects with visual markers for more accurate error analysis and correction in the distractor. **ii)** When comparing the results of PEIFG w/o VMP, PEIFG w/o CLIP with PEIFG respectively, it becomes evident that both visual branches (i.e., VMP and CLIP image encoder) for two types of visual features contribute to the performance improvement. **iii)** When applying expert prompts to PEIFG, we obtain a significant improvement in feedback and distractor generation. Specifically, the performance increases from 15.15 to 17.69 and 39.64 to 41.08 on BLEU-4 and ROUGE$_L$ metrics respectively. This outcome indicates the success of the expert prompt selection strategy and how expert prompts effectively aid the LLM in understanding and correcting errors to generate corresponding feedback. **iv)** Finally, we investigate the impact of the refinement operation assisted by GPT-4 on feedback generation. After refinement, we find this step can improve the performance on all metrics, which further ensures the logical coherence between the feedback and input information. In particular, the prediction accuracy of educational levels is improved from 86% to 88%.

## 5.5 Case Study

Fig. 3 shows the feedback generated by VisualGLM, CogAgent and our model PEIFG. Intuitively, PEIFG generates more grounded feedback compared to other baseline models. Specifically, we find that: **i)** The feedback generated by VisualGLM frequently lacks crucial information in its explanations that would lead to the correct answer. Specifically, as shown in Fig. 3, the explanation generated by VisualGLM incorrectly classifies the educational level and recognizes the location of person9 as being 'in the ring' rather than the actual "by the ring". Worse still, it merely states the reason for the error without further elaborating on the correct answer that "person9 is a commentator". In contrast, both CogAgent and our PEIFG accurately recognize the location of "person9" and correctly conclude that "person9 is a commentator". **ii)** Compared our PEIFG with CogAgent, the explanations generated by CogAgent overlook the visual clue 'mic', which may hinder to logically deduce the correct answer that 'person9 is a commentator'. Although CogAgent contains eight times the number of parameters as our PEIFG model, our model still manages to more accurately capture visual clues "by the ring" and "mic" for error correction. This can be attributed to the efficacy of our expert prompts, which effectively instruct the LLM to focus more on visual clues relevant to error correction.

## 6 Conclusion

In this paper, we present a pioneering work to investigate the error correction capabilities in visual commonsense reasoning of existing large multimodal models (LMMs), noted for their exceptional reasoning skills. Therefore, we leverage GPT-4 to construct the VCR-DF benchmark for evaluation of error correction by feedback generation task. Additionally, we propose the PEIFG model to identify visual commonsense errors and provide explainable feedback. Specifically, we first design a visual feature extractor to obtain the integrated visual features. Subsequently, the integrated visual features and language instruction are jointly used to select the relevant

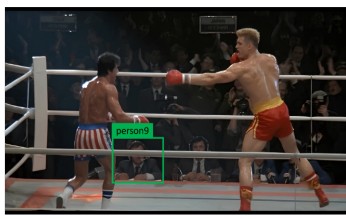

**Question**: Why is person9 by the ring?

**Answer**: He is one of the commentators for the match.

**Distractor**: Person9 is a coach for person0.

| | |
|---|---|
| **Ground Truth** | **Educational level**: Analyze
**Error location**: Misidentifying person9's role.
**Explanation**: This distractor confuses the roles within the boxing match, incorrectly assuming person9 is a coach based on his proximity to the ring, rather than a commentator as indicated by the context of wearing a headset with a mic. |
| **VisualGLM** | **Educational level**: **Understand**
**Error location**: Assumption of a coaching role.
**Explanation**: The error arises from assuming **person9's presence in the ring** means he is a coach, rather being an observer for the match. |
| **CogAgent** | **Educational level**: Analyze
**Error location**: Misinterpretation of the role of coach.
**Explanation**: The error arises from a misunderstanding that the one close to the ring is the coach. This is a misconception that person9 is a coach, but not a commentator. |
| **PEIFG** | **Educational level**: Analyze
**Error location**: Incorrect assumption of a professional role.
**Explanation**: This distractor doesn't focus on person9 wearing a headset with a mic, but only infers that it is a coach based on it by the ring, rather than a commentator based on more details. |

**Figure 3: Case study of the generated feedback by VisualGLM, CogAgent and our model PEIFG.**

expert prompts from the pool. Finally, we effectively integrate the visual features and learnable expert prompts into multimodal instruction and design a refinement strategy for feedback generation. The experimental results indicate that our PEIFG outperforms existing LMMs. We hope this learning to correction process would chart a new direction to evaluate the capabilities of LMMs.

## Acknowledgments

This research is supported by the National Natural Science Foundation of China (62076100), the Fundamental Research Funds for the Central Universities, South China University of Technology (x2rjD2240100), the Science and Technology Planning Project of Guangdong Province (2020B0101100002), Guangdong Provincial Fund for Basic and Applied Basic Research—Regional Joint Fund Project (Key Project) (2023B1515120078), Guangdong Provincial Natural Science Foundation for Outstanding Youth Team Project (2024B1515040010), the China Computer Federation (CCF)-Zhipu AI Large Model Fund, the Hong Kong Polytechnic University's Postdoc Matching Fund (project no. P0049003).

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
