# OpenReview forum: "Learning to Correction: Explainable Feedback Generation for Visual Commonsense Reasoning Distractor"
_acmmm.org/ACMMM/2024/Conference — MM2024 Poster_

### Official Review · Reviewer_PzzN · 2024-05-23

**Rating:** 3
**Confidence:** 3

**Summary:**

This paper presents to address the ```distractor`` problem in visual commonsense reasoning.
In this vein, this work makes the following two contributions:
First, the authors collect a new dataset based on the original VCR dataset.
The dataset curation leverages the generation of GPT-4.
Second, the authors propose a method to address this problem.
The experiments on this new dataset prove the effectiveness of the proposed method.

**Strengths:**

- The authors propose to leverage GPT-4 to generate distracted sentences, which is interesting and novel.
- This paper contains extensive experiments on the new dataset especially shown in Table 1.
We can see the authors re-implement so many baselines and produce their results on this new dataset.

**Limitations:**

- I'm confused about the key definition of this task.
It seems we do not have to limit this problem to visual commonsense reasoning.
In particular, the dataset itself has some flaws and does not involve much ```visual commonsense```.
As a result, the generated distractors do not demonstrate this virtue as well.
- The proposed method is too complicated to understand.
It leads to two serious problems:
1) it is hard for readers to distinguish which parts are novel and are designed by this work.
2) It is almost impossible to reproduce the results of this work.
- The authors leverage a framework from the Pedagogical domain.
To me, this will 'distract' the key contribution of this work as it generally introduces many difficulties to grasp the key idea and there is scanty connection between this work and Pedagogy.

**Suitability:**

2

---

### Official Review · Reviewer_jVKk · 2024-05-25

**Rating:** 5
**Confidence:** 3

**Summary:**

This paper introduces a feedback generation task for the visual commonsense reasoning distractors to evaluate the error correction ability of large multimodal models (LMMs). In particular, the paper constructs the VCR-DF dataset including new distractors and explainable feedback, which can serve as a benchmark to evaluate the error correction ability of LMMs. Besides, the paper devises a pedagogical expert instructed feedback generation model to conduct the feedback generation. Extensive experiments are conducted to verify the effectiveness of the proposed model.

**Strengths:**

Strong points:
1. The paper aims to investigate the error correction ability of LMMs, which is vital and promising.
2. The paper provides a comprehensive survey of related studies.
3. The paper provides figures to show the proposed model, which is vivid and can facilitate understanding.

**Limitations:**

Weak points:
1. The paper utilizes both the region-level and global-level features, but does not conduct ablation studies to verify the necessity.
2. The paper is suggested to provide a more detailed description of the VCR-DF dataset.
3. I wonder if the constructed VCR-DF dataset can be released to facilitate other researchers.

**Suitability:**

3

---

### Official Review · Reviewer_pGkx · 2024-05-26

**Rating:** 4
**Confidence:** 3

**Summary:**

This paper aimed to explore the ability of Large Multimodal Models (LMMs) in correcting potential visual commonsense errors in distractor and proposed a dataset VCR-DF to tackle this.

**Strengths:**

1, This paper constructed the VCR-DF dataset to evaluate the ability of LMMs in clarifying the reasons behind errors in VCR distractors.
2, To engage the LMM with strong reasoning abilities in error correction, they propose the Pedagogical Expert Instructed Feedback Generation (PEIFG) mode.
3, Experimental results demonstrate that the paper proposed Pedagogical Expert Instructed Feedback Generation (PEIFG) model significantly outperforms existing LMMs.

**Limitations:**

1, What is the difference of the task and asking the model to describe why the correct answer was chosen?
2, In the methods section, the correct answer is provided to the model, and then generates feedback for distractor. Could the introduction of correct answer offer additional information? If the correct answer is absent, would the model still maintain good performance?
3, Figure 2 is unclear, especially regarding the VFE. I don’t understand why only Stage 1 training is shown. Where is Stage 2?
4, All the grounding feedbacks in VCR-DF are generated by GPT-4. Why are the results of the proposed model still higher than those of GPT-4 in Table 2?

**Suitability:**

2

---

### Official Review · Reviewer_Tjci · 2024-05-26

**Rating:** 3
**Confidence:** 3

**Summary:**

This article introduces a novel approach to enhance Large Multimodal Models' (LMMs) capabilities in correcting visual commonsense reasoning (VCR) errors within distractors, inspired by the method a human teacher uses to craft distractors for testing and educating students. The article pioneers the use of GPT-4 as a "teacher" to compile an explainable feedback dataset named VCR-DF, aimed at benchmarking LMMs' abilities to identify and explain errors in VCR distractors, leading to the correct answers. Furthermore, the research proposes the Pedagogical Expert Instructed Feedback Generation (PEIFG) model, which integrates learnable expert prompts and multimodal instructions to guide feedback generation. The experimental findings indicate that PEIFG significantly surpasses existing LMMs in performance, suggesting a new direction for evaluating LMMs' capabilities.

**Strengths:**

This article presents several strengths in its approach to enhancing the error correction capabilities of large multimodal models (LMMs) through the proposed Pedagogical Expert Instructed Feedback Generation (PEIFG) framework.
1.Innovative Integration of Visual and Textual Feedback: The PEIFG model introduces a novel method for combining visual features extracted using a visual feature extractor (VFE), which includes a Visual Marker Perceiver (VMP) and CLIP image encoder.
2.Customized Expert Prompt Selection for Enhanced Error Correction: The development of an Expert Prompt Selector (EPS) within the PEIFG model represents a significant advancement. By selecting the most relevant expert prompts from a learnable prompt pool, the model can provide specialized guidance for feedback generation.
3. Refinement and Evaluation through Reinforcement Learning: The inclusion of a refinement step that employs reinforcement learning to improve the explainability and faithfulness of the generated feedback is a crucial advantage.

**Limitations:**

1.	The different representations of VMP in Figure 2 may cause confusion. (Training or Freezing)
2.	Why train both VPM and OPT models simultaneously? Is it necessary to train the OPT model?
3.	The Expert Prompt Selection needs more theoretical explanation. At the same time, more detailed ablation experiments are needed to analyze the role of the internal components of this module, such as the ablation experiment of top_k.
4.	The section title "5.4.1 performance Comparison" should be changed to "5.4.1 Performance Comparison".

**Suitability:**

3

---

### Meta-Review · Area_Chair_LRps · 2024-06-28

**Recommendation:** Accept (Poster)
**Confidence:** 5

**Metareview:**

Overall, this paper is innovative with abundant experimental content. The authors have effectively addressed the reviewers' questions, and we agree to accept this paper.